# Assessing alternative indicators for Covid-19 policy evaluation, with a counterfactual for Sweden

Chiara Latour[1], Franco Peracchi[2], Giancarlo Spagnolo[1,2]*

**1** University of Stockholm, Stockholm, Sweden, **2** University of Rome Tor Vergata and EIEF, Rome, Italy

☯ These authors contributed equally to this work.
* spagnologianca@gmail.com

**Data Availability Statement:** The data underlying the results presented in the study are available from Our World in Data (https://ourworldindata.org/coronavirus-data.) and the European CDC (https://www.ecdc.europa.eu/en/publications-data/

## Abstract

Using the synthetic control method, we construct counterfactuals for what would have happened if Sweden had imposed a lockdown during the first wave of the COVID-19 epidemic. We consider eight different indicators, including a novel one that we construct by adjusting recorded daily COVID-19 deaths to account for weakly excess mortality. Correcting for data problems and re-optimizing the synthetic control for each indicator, we find that a lockdown would have had sizable effects within one week. The much longer delay estimated by two previous studies focusing on the number of positives cases is mainly driven by the extremely low testing frequency that prevailed in Sweden in the first months of the epidemic. This result appears relevant for choosing the timing of future lockdowns and highlights the importance of looking at several indicators to derive robust conclusions. We also find that our novel indicator is effective in correcting errors in the COVID-19 deaths series and that the quantitative effects of the lockdown are stronger than previously estimated.

## 1 Introduction

In this paper we argue that focusing attention on a single "best" indicator of the COVID-19 epidemic (e.g., the recorded number of infections or the recorded number of deaths) in order to assess the potential effects of a containment policy, as often done in the literature and in policymaking, may be quite misleading. However, if several alternative indicators point in the same direction and produce qualitatively similar results, this provides a much stronger basis for policy evaluation.

We illustrate our argument by estimating the potential effects of a lockdown using Sweden as the reference country. There are two reasons why this is an interesting country to consider. First, Sweden plays an important role in the international debate because it chose a mitigation strategy characterized by much weaker restrictions than most other comparable countries. For this reason, it represents a unique benchmark against which to evaluate the potential effects of a lockdown. Second, Sweden was the subject of several recent studies, focusing mainly on the number of recorded COVID-19 infections, which we replicate using a range of alternative indicators. In particular, we revisit the recent studies of Born et al. [1] and Cho [2] who employ the synthetic control method introduced by Abadie and Gardeazabal [3] to create a

**Funding:** The author(s) received no specific funding for this work.

**Competing interests:** The authors declare no competing interests for this work.

counterfactual for Sweden and thereby estimate what would have happened to the number of recorded infections if Sweden had introduced a lockdown. For comparability with these studies, we focus on the first wave of the epidemic (February–June 2020). Another reason for restricting attention to the first wave is the small variation of the starting date of the lockdown in the countries that adopted this policy in Spring 2020, which lends credibility to the "ceteris paribus" assumption needed by the methodology. A third reason is that the analysis of subsequent waves (Fall 2020–Summer 2021) is complicated by the spread of mutated and more infective variants of the virus.

While trying to maintain a similar approach to Born et al. [1] and Cho [2] to ensure comparability, our analysis departs in some important respects from these studies. First, in addition to recorded COVID-19 infections and deaths, which are likely to underestimate actual infections and deaths, we consider two other important outcomes: i) adjusted COVID-19 deaths–a measure we construct to reconcile the series of daily recorded COVID-19 deaths with the series of weekly excess deaths, which many view as a more reliable indicator because of the wide cross-country differences in both the intensity of testing and the recording of COVID-19 deaths; and ii) the ratio between the number of recorded infection and the number of tests, or positive rate, which is important because testing intensity changed dramatically in Sweden during our sample period. Second, for all outcomes, we consider both cumulated and daily values, resulting in a total of eight different indicators. We also address several problems in the data that had not been previously identified.

All our indicators suggest that a lockdown would have had a strong effect in reducing the impact of the epidemic in Sweden. Most importantly for the design of future containment policies, we find that a lockdown would have displayed its effects within a week from its introduction. This finding contrasts with the much longer lags (three to five weeks) in the effect of a lockdown estimated in Born et al. [1] and Cho [2] who focused mainly on recorded cumulative infections, but is consistent with other studies focusing on different countries or using a different methodology (e.g., [4–6]).

Since the time lag with which a lockdown displays its effects is of primary importance for the design and timing of containment policies, we dig further into the sources of these conflicting results. Comparing the cumulative positive rate with our daily indicators allows us to conclude that the much longer delay obtained by focusing on cumulated recorded infections reflects more the very low intensity of testing that prevailed in Sweden in the first months of the epidemic than the slow adjustment typical of cumulated measures. Taken together, our results highlight the importance of using multiple indicators to obtain robust policy conclusions.

As for the magnitude of the potential effects of a lockdown in Sweden, after correcting for data issues we obtain estimates that are larger than those presented in Born et al. [1] and Cho [2].

The rest of the paper is organized as follows. Section 2 provides examples of the wide cross-country heterogeneity in recording COVID-19 deaths and testing intensity, and how the latter changed over time, especially in Sweden. Section 3 describes our methodology. Section 4 presents our data and the new measure of COVID-19 mortality introduced in the present study. Section 5 presents our results and provides some robustness checks. Section 6 discusses our results and concludes.

## 2 Heterogeneity in death recording and testing policies

The reason why we believe it is crucial to consider additional outcomes is the presence of large differences across countries in the way COVID-19 deaths are recorded and in the intensity of

testing, as well as within-country changes in testing intensity during the relevant period, most crucially in Sweden. This section provides examples and simple quantifications of these forms of spatial and temporal heterogeneity.

## 2.1 Heterogeneity across countries

The procedures for ascribing deaths to COVID-19 differ across countries, both before and after April 16, 2020, when the WHO issued its guidelines [7]. The procedure recommended by the WHO, and adopted by countries such as Belgium, Canada, France, Germany, and Greece, uses clinically confirmed or probable COVID-19 cases and does not depend on the availability of a laboratory test. An alternative definition of COVID-19 deaths, adopted by countries such as Austria, Italy, the Netherlands, Spain, and the United Kingdom, relies instead primarily on a positive laboratory test.

Countries following the WHO recommendations are likely to capture a greater share of the deaths caused by COVID-19. Even among these countries, however, recording of the cause of death can vary because of different practical implementations of the proposed guidelines, different criteria for death certification, and different coding practices. For example, some countries still require a positive test result (e.g. Greece), while others (e.g. Canada) include anybody with a COVID-19 diagnosis, even if death was triggered by something different from the virus (e.g. trauma). Guidelines may also change over time. From what we understand, Sweden followed the WHO recommendations but was rather generous in ascribing deaths to COVID-19 [8].

Regarding cross-country heterogeneity in the intensity of testing, Fig 1 reports the average number of tests per 100,000 inhabitants in 12 European countries between March 15 and May 20, 2020. The countries considered are those included in our baseline analysis in Section 5.2.

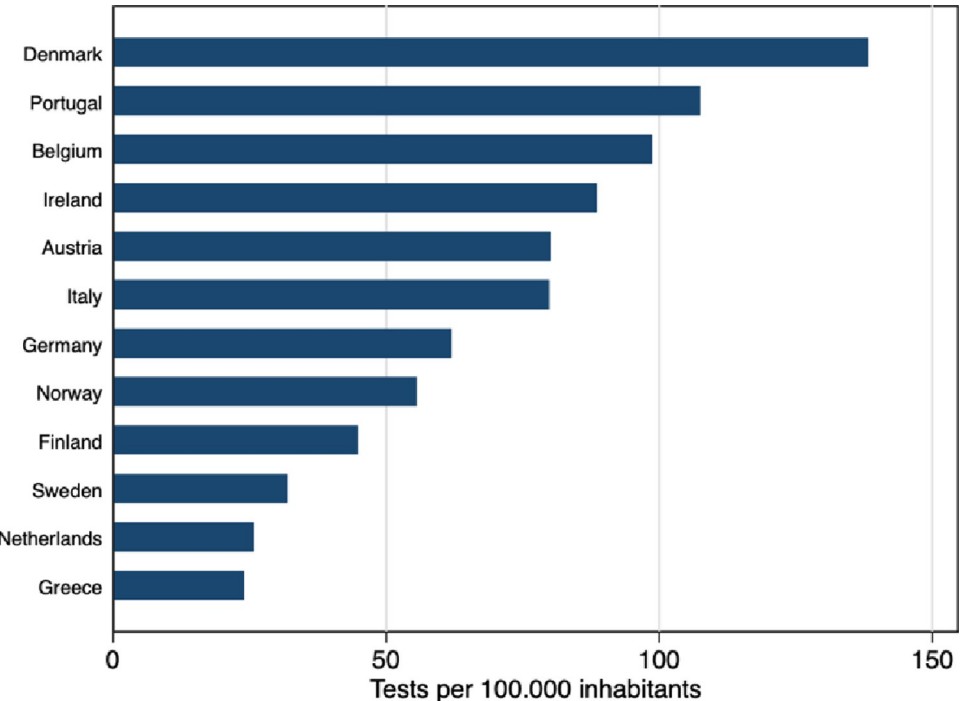

**Fig 1. Number of tests in 12 European countries.** *Notes*: The figure shows the average number of tests per 100,000 inhabitants in 12 European countries between March 15 and May 30. Data source: ECDC.

Testing intensity has been varying wildly during the period considered, with Sweden being one of the countries that tested less.

## 2.2 Changes in testing policy in Sweden

While remaining a laggard in terms of the number of performed tests for the large part of our sample period, Sweden tried to step up its testing capacity over time.

Prior to March 12, 2020, the Swedish strategy was to test all people who had been in areas considered at high risk of infection, like China, Northern Italy or Austria. However, due to shortages in testing equipment, this strategy was rapidly changed, and testing only targeted medical care staff and people with heavy symptoms and in need of hospitalization [9]. Sweden then reversed this policy announcing on March 31, 2020, a plan to expand testing capacity to all critical services with the aim of carrying out 100,000 tests per week [10].

Although the Swedish government did not manage to rapidly step up its testing capacity, this policy change resulted in a steady increase in the number of tests performed each week. As shown in Fig 2, the expansion of testing was initially accompanied by a faster increase in the number of recorded infections that has little to do with the dynamics of the infection in the country, resulting in a rapid rise in the positive rate. This is not surprising since the people who first asked to be tested when more testing became available were likely those worried about their infection status but unable to be tested before, hence exhibited a higher probability to have contracted the virus. In the second half of April 2020, the positive rate dropped at a lower level and started to decline slowly. Finally, on June 4, 2020, the Swedish government managed to implement a large expansion of its testing capacity and offered free testing to all citizens.

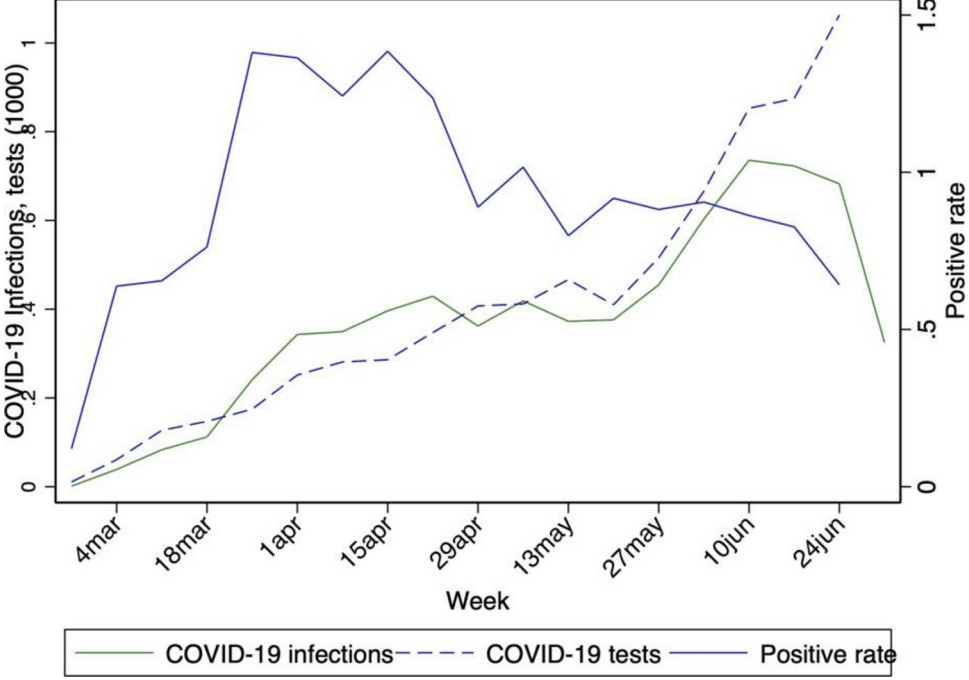

**Fig 2. Weekly number of infections and weekly number of tests in Sweden.** *Notes*: The figure shows the weekly number of infections, the weekly number of tests, and the positive rate in Sweden up to June 22. Data source: ECDC and Our World in Data.

As an example of how misleading it can be to rely upon a single indicator (in this case, the number of recorded infections), a few days after Sweden finally managed to step up significantly its testing capacity while the positive rate was starting to fall, the WHO included Sweden in a set of 11 European countries with "accelerated transmission that if left unchecked will push health systems to the brink once again" [11]. The Public Health Agency of Sweden rightly rejected this statement arguing that the Swedish testing policy changed dramatically in June 2020 and for that reason, the data on the number of infections had been misinterpreted by the WHO.

## 3 Methodology

We employ the same methodology as Born et al. [1] and Cho [2] namely the synthetic control method introduced by Abadie and Gardeazabal [3]. This is done both to ensure comparability and because of the simple and intuitive nature of this increasingly popular method. In this paper, we refer to the slightly older version of Born et al [1], but their main results for the period relevant to our paper did not change in the recently published version [12]. After briefly describing the method, we summarize its key elements. We refer to Abadie [13] for a thorough review.

### 3.1 The synthetic control method

This method estimates the time-varying effect of a "treatment" (an intervention or policy) on some outcome of interest for a specific "treated unit" (an administrative district, geographical region, or country) by the difference in the time path of the outcome between the treated unit after the treatment and an artificial or "synthetic" unit constructed by taking a weighted average of a suitably selected set of untreated units (the "donor pool"). The weights given to the units in the donor pool are nonnegative, sum to one, and are chosen to minimize the distance between the treated and the synthetic unit in a space of unit-specific indicators that may include pre-treatment values of the outcome of interest. In practice, these weights are usually "sparse", that is, only a few units receive positive weights. When only one donor unit receives a positive weight, the method reduces to the simple difference between two units.

As argued by Abadie [13], "the synthetic control method is based on the idea that, when the units of observation are a small number of aggregate entities, a combination of unaffected units often provides a more appropriate comparison than any single unaffected unit alone." The method generalizes comparative case studies by formalizing the choice of the comparison units and the criteria for the comparison.

Notice that, unlike the vast literature on treatment effects, the synthetic control method estimates a time-varying individual treatment effect, not the mean or a quantile of the distribution of individual treatment effects.

### 3.2 Key elements of the method

The key elements of the synthetic control method are: (i) the choice of treatment (in our case, the decision to not impose a nationwide lockdown in March 2020), (ii) the choice of the treated unit (in our case, Sweden), (iii) the choice of the outcome of interest (in our case, any of the indicators discussed in Sections 4.1–4.3), (iv) the length $T_0$ of the pre-treatment period (discussed in Section 5.1), (v) the choice of the "donor pool" (in our case, the set of countries to which Sweden is compared, also discussed in Section 5.1), (vi) the choice of unit-specific characteristics (discussed in Section 4.4), and (vii) the choice of metric to measure distance in the space of unit-specific characteristics (in our case, the same as Born et al. [1] and Cho [2]).

Abadie [13] argues that "the ability of a synthetic control to reproduce the trajectory of the outcome variable for the treated unit over an extended period of time [. . .] provides an indication of low bias", that "the risk of overfitting may also increase with the size of the donor pool, especially when $T_0$ is small", and that "each of the units in the donor pool have to be chosen judiciously to provide a reasonable control for the treated unit. Including in the donor pool units that are regarded by the analyst to be unsuitable controls [...] is a recipe for bias". Further, "the credibility of a synthetic control estimator depends on its ability to track the trajectory of the outcome variable for the treated unit for an extended pre-intervention period."

In practice, results from the synthetic control method tend to be quite sensitive to the choices made regarding all the elements listed above. Abadie [13] recommends choosing a donor pool that is not too large, with units that are not too different in terms of both observable and unobservable characteristics. He also recommends choosing a pre-treatment period that is not too short. Since these choices remain largely "ad hoc", we rely on various robustness checks that are presented in Section 5.3.

## 4 Data

This section presents our data and the new measure of COVID-19 deaths introduced in the present study.

### 4.1 COVID-19 infections and deaths

The daily and cumulative series of recorded COVID-19 infections and deaths are taken from the Coronavirus Pandemic section of *Our World in Data* [14], which collects data on confirmed COVID-19 infections and deaths originally published by the European Centre for Disease Prevention and Control (ECDC). By recorded daily infections we mean the number of recorded new cases in a given day and by cumulative recorded infections we mean the running sum of recorded new cases from the start of the epidemic until that day (with missing values treated as zeros). Recorded daily and cumulative deaths are similarly defined. All these data are available at daily frequency for all countries considered. To ensure comparability across countries, we normalize all values dividing by the estimated population size of a country at the beginning of the year 2020 and convert to cases per 1 thousand inhabitants. Notice that recorded infections are lower than actual infections for reasons that include the absence of random testing, problems of missing data, and imperfect test accuracy [15, 16].

Daily series of recorded infections and deaths are subject to strong day-of-the-week effects. They display some small negative values for several countries and a few very large positive or negative values for two countries, France and Spain. The presence of implausibly large positive values or inadmissible negative values in the daily series reflects periodic adjustments by the agencies issuing the data, whose nature, magnitude, and frequency vary both across countries and over time. These problems appear not to have been identified in previous studies based on the cumulative version of these data. In particular, the negative values in the daily number of recorded infections and deaths imply declines in the cumulative values which may significantly affect the results of the synthetic control method.

To reduce the impact of these data anomalies and control for day-of-the-week effects, we smooth the original series by taking 7-day moving averages. This does a good job in reducing the noise in the data for all countries considered, except France and Spain where the outliers are just too large. Because of this, we think the best course of action is to drop these two countries from the donor pool, though we add them again in one of the robustness analyses in Section 5.3.

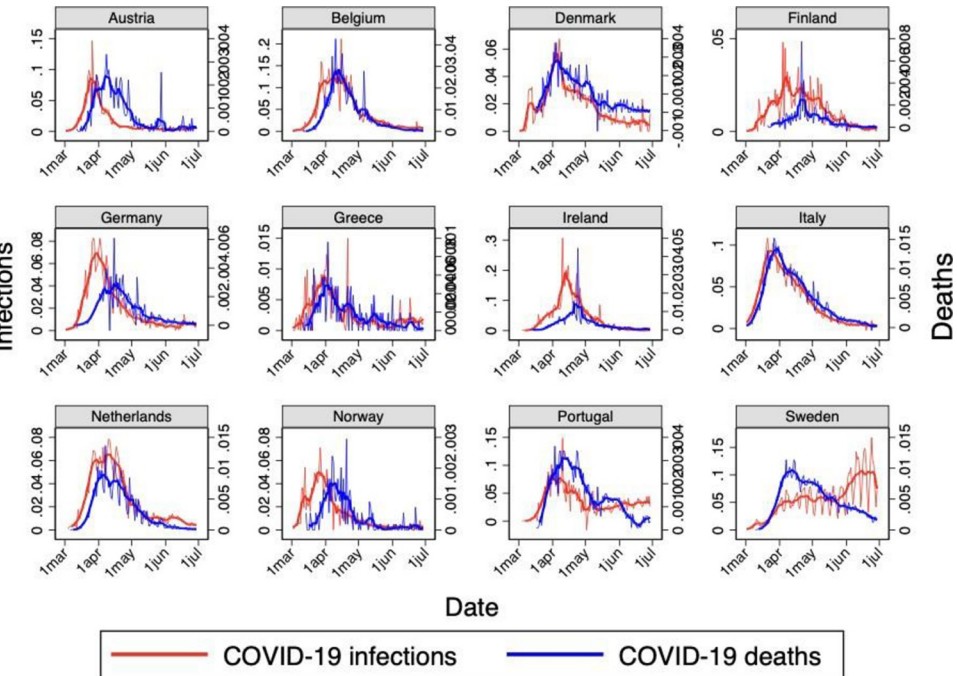

**Fig 3. Recorded daily COVID-19 infections and deaths.** *Notes*: The figure shows the number of daily COVID-19 infections and deaths per 1,000 inhabitants. The thinner profiles are the original daily series, while the thicker profiles are the smoothed series obtained by taking 7-day moving averages. Data source: Our World in Data.

Fig 3 compares the original and the smoothed daily series (respectively the thinner and the thicker lines) of recorded COVID-19 infections and deaths in Sweden and 11 other European countries, namely those considered by Born et al [1] with the exception of France and Spain. While the profile of recorded daily infections is quite different for Sweden, due to the mentioned changes in its testing policy, the profile of recorded daily deaths is qualitatively similar in all countries considered, except for the much higher force of mortality in Belgium, Italy, the Netherlands, and Sweden.

In addition to the number of recorded COVID-19 infections per inhabitant, we also consider the positive rate, namely the ratio between the number of recorded COVID-19 infections and the number of COVID-19 tests. Testing data are available for all countries considered except the Czech Republic and Spain. When available, daily data on the number of new COVID-19 tests have been downloaded from the website of *Our World in Data* [14]. When daily data are not available (as in the case of Croatia, Germany, Greece, Netherlands, Poland, and Sweden), we use weekly data downloaded from the website of the ECDC and then construct a daily series by linear interpolation [17].

## 4.2 Mortality and excess mortality

Data on mortality from all causes are taken from the website of the *Financial Times* [18]. These data are only available at the weekly level and are unavailable for Ireland and Romania.

Excess mortality is defined as the number of deaths recorded in a given period on top and beyond what we would have expected given mortality in the recent past. Operationally, it is computed as the difference between mortality in 2020 and average mortality in the 5-year period between 2015 and 2019. Although excess mortality is only available on a weekly basis, we use this information to construct a simple correction of the daily series of recorded

COVID-19 death to match the weekly number of excess deaths. We describe this correction in the next section.

Ritchie et al. [14] and Krelle et al. [19] among others, argue that excess mortality is more comparable across countries because it is less sensitive to structural differences, such as the efficiency of the health care system, or to demographic characteristics, such as the distribution of the population by age. They argue that excess mortality is also a better measure for policy analysis because it avoids miscounting from under-reporting of COVID-19 related deaths or from other health conditions left untreated because of the epidemic. In fact, during an epidemic, we might have an increase in the number of deaths from other unrelated causes because hospitals are overwhelmed and work at full capacity, which leads to many conditions being left untreated or many people not seeking treatment. At the same time, however, there might be fewer deaths from other causes such as road accidents given the mobility restrictions.

Some important points that could affect data comparability across countries must also be kept in mind. First, the accuracy of raw mortality data can vary across countries due to differences in the death registration system. Second, due to lags in registration, death counts by week of registration may not reflect the actual time profile of mortality. Lastly, when using excess deaths per capita, countries with an older population will tend to have higher normal death rates, so caution is needed when comparing per capita excess mortality across countries with different population structures.

### 4.3 Using excess mortality to estimate total COVID-19 deaths

The number of recorded COVID-19 deaths is likely to represent a downward biased estimate of the death toll caused by the disease [20]. The bias varies across countries and over time because of differences in both the testing policies and the procedures for attributing deaths to COVID-19. In this section we propose a simple way of accounting for unrecorded COVID-19 related deaths, that is, deaths not attributed to COVID-19, by making use of the available weekly data on excess mortality.

Let $T_{dj}$ denote the observable number of total deaths (i.e. deaths from all causes) on day $d = 1, 2, \ldots, 7$ of week $j$ of 2020, and let $T_{dj}^0$ denote the average number of total deaths on day $d$ of week $j$ during the baseline period 2015–2019. Excess mortality in week $j$ of 2020 is measured by the difference $\overline{T_j} - \overline{T_j^0}$, where $\overline{T_j}$ and $\overline{T_j^0}$ are weekly averages of $T_{dj}$ and $T_{dj}^0$ respectively. We define excess deaths in week $j$ of 2020 as the difference $\overline{T_j} - \overline{T_j^0}$. This is negative when $\overline{T_j} < \overline{T_j^0}$, as for most countries at the beginning of 2020 and again during the Summer of 2020.

Under the assumption that COVID-19 is the only important cause of higher mortality in 2020 relative to the baseline, the positive part of excess deaths, namely $E_j = \max\{0, \overline{T_j} - \overline{T_j^0}\}$ is a measure of total (recorded and unrecorded) daily COVID-19 related deaths in week $j$ of 2020. If $Y_{dj}$ denotes the smoothed number of recorded COVID-19 deaths on day $d$ of week $j$ of 2020, obtained by taking a 7-day moving average of recorded daily COVID-19 deaths, the average daily number of unrecorded COVID-19 deaths in week $j$ of 2020 is measured by

$$Z_j = E_j - \overline{Y}_j,$$

where $\overline{Y}_j = \sum_{d=1}^{7} Y_{dj}/7$. We can then estimate the smoother number of unrecorded COVID-19 deaths on day $d$ of week $j$ of 2020 by linear interpolation,

$$\hat{Z}_{dj} = \frac{d}{7}Z_j + \left(1 - \frac{d}{7}\right)Z_{j-1}.$$

Adding the result to the smoothed daily number of recorded COVID-19 deaths gives the following estimate of the daily number of total COVID-related deaths

$$\hat{X}_{dj} = Y_{dj} + \hat{Z}_{dj}.$$

We shall refer to $\hat{X}_{dj}$ as adjusted COVID-19 deaths. The adjustment is sizable in countries, such as the Netherlands and Sweden, where excess mortality in Spring 2020 was positive and large.

### 4.4 Country characteristics

In constructing the synthetic control for Sweden, we initially consider the same set of country characteristics employed by Born et al. [1], namely population size and the share of urban population. In one of the robustness analyses in Section 5.3, we expand this set by adding household size (also considered by Cho [2]), GDP per capita, median population age, the fraction of people aged 70+, the number of hospital beds per inhabitants, and life expectancy at birth. All country characteristics are measured as of the latest available year. Urban population data are taken from the *World Bank* and data on all other characteristics are from *Our World in Data* [14].

## 5 Results

After discussing in Section 5.1 the details of our implementation of the synthetic control method, Section 5.2 presents the results from our baseline case. Results from a number of robustness checks are briefly discussed in Section 5.3.

### 5.1 Implementation details

We follow Born et al. [1] for the choice of the donor pool and the set of country characteristics considered, but we exclude France and Spain for the reasons discussed in Section 4.1. Thus, our donor pool consists of 11 countries: 10 Western European Union countries with more than 1 million inhabitants (Austria, Belgium, Denmark, Finland, Germany, Greece, Ireland, Italy, Netherlands, and Portugal) plus Norway. Compared to [2], this gives a smaller but more homogeneous donor pool. In one of the robustness checks in Section 5.3, we examine the effect of broadening the donor pool by including most of the countries considered by Cho [2], the exceptions being non-European countries and countries with less than 1 million inhabitants. The set of country characteristics only includes population size and the share of urban population. In the last robustness check in Section 5.3 we enlarge this set to include several other socio-demographic indicators.

Unlike Born et al. [1], and more in line with Cho [2], we extend the length of the post-lockdown period till the end of June 2020 to fully allow for the sharp increase in testing rates that occurred in Sweden after an initial period of very low testing (see Section 2.2) to fully display its effects.

Most importantly, as already mentioned, we expand the set of outcomes considered relative to Born et al. [1] and Cho [2]. In addition to the number of recorded COVID-19 infections and deaths, we also include the number of adjusted COVID-19 deaths (constructed as described in Section 4.3) and the positive rate (computed as the ratio between the number of recorded infections and the number of tests performed). Since changes in testing policy directly and strongly affect the number of recorded new cases, but do not necessarily affect their ratio to the number of tests performed, the positive rate is a very informative outcome in our context.

In addition to cumulative indicators, which look very smooth since positive and negative deviations from the trend tend to offset each other, we also consider smoothed daily indicators. This is because we are interested in how fast the effect of a lockdown would have kicked in, and cumulative outcomes naturally "hide" for some time the effects of a policy. Further, unlike Cho [2], the weights assigned by the synthetic control method to the countries in the donor pool are not kept constant but are re-optimized for each of the eight indicators considered.

Our indicators are not orthogonal, but pairwise correlations vary a lot by country. The correlation between the smoothed daily indicators is mostly positive, with correlation coefficients that range from less than .5 to over .95 depending on the country and the indicators considered. A few negative correlations are also observed. Negative coefficients are more common when considering the correlation between cumulative indicators, and between daily and cumulative indicators.

For the countries in the donor pool, we take the pre-lockdown period to consist of the 2 weeks before the start date of the lockdown (13 days in the case of the positive rate). To improve comparability across countries, we transform time in deviations from the "treatment date", so day 0 is when the lockdown was introduced. For Sweden, that never adopted a lockdown, day 0 is set to March 17, the mean start date of the lockdown in the donor pool. As with the other papers cited, we ignore cross-country differences in the characteristics and intensity of the lockdown.

The frequency distribution of the incubation period for COVID-19 –i.e., the time between exposure to the virus and symptom onset–has a median of 7 days [21], while the median length of time from symptom onset to death ranges between 17 and 19 days [22, 23]. Thus, we take 7 days as the average length of the incubation period and 18 days as the average length of time from symptom onset to death. When using COVID-19 deaths and adjusted COVID-19 deaths, we shift the treatment date by 18 days from the lockdown date to account for the expected time between symptom onset and death.

## 5.2 Sweden vs. synthetic Sweden

### 5.2.1 COVID-19 deaths vs. adjusted COVID-19 deaths.   Fig 4 shows the profile of Sweden versus synthetic Sweden for cumulative COVID-19 deaths and cumulative adjusted COVID-19 deaths over our sample period, which extends for 105 days after the lockdown date ending with June 30, 2020.

There is one evident difference between the two indicators. While the profile of cumulative adjusted COVID-19 deaths for Sweden never drops below its synthetic counterpart, cumulative COVID-19 deaths between days 0 and 15 are slightly smaller than for synthetic Sweden. This difference could be associated with anomalies in recorded COVID-19 deaths in early April when, on April 4 (day 18), Sweden reported a negative and relatively large number of deaths, most likely with the intent to correct over-reporting in previous days.

As mentioned in Section 5.1, we use an 18-day lag to account for the time between symptoms onset and death. However, the range of the lags between infections and deaths estimated in previous studies is very wide. For example, Rees et al. [24] report that the estimated median from 52 papers for the length of stay in the hospital amongst patients who died ranges between 4 and 21 days, while Faes et al. [25] estimate that the median length of time between symptoms onset and hospitalization ranges between 3 and 10.4 days depending on individual-specific characteristics. Consequently, the median length of time from symptom onset to death can be substantially wider than the 18-days that we assume. We therefore carry out a sensitivity analysis on the lag selection by shifting the treatment date one week further. The results are shown in Fig 5. While the behavior of COVID-19 deaths changes substantially compared to before,

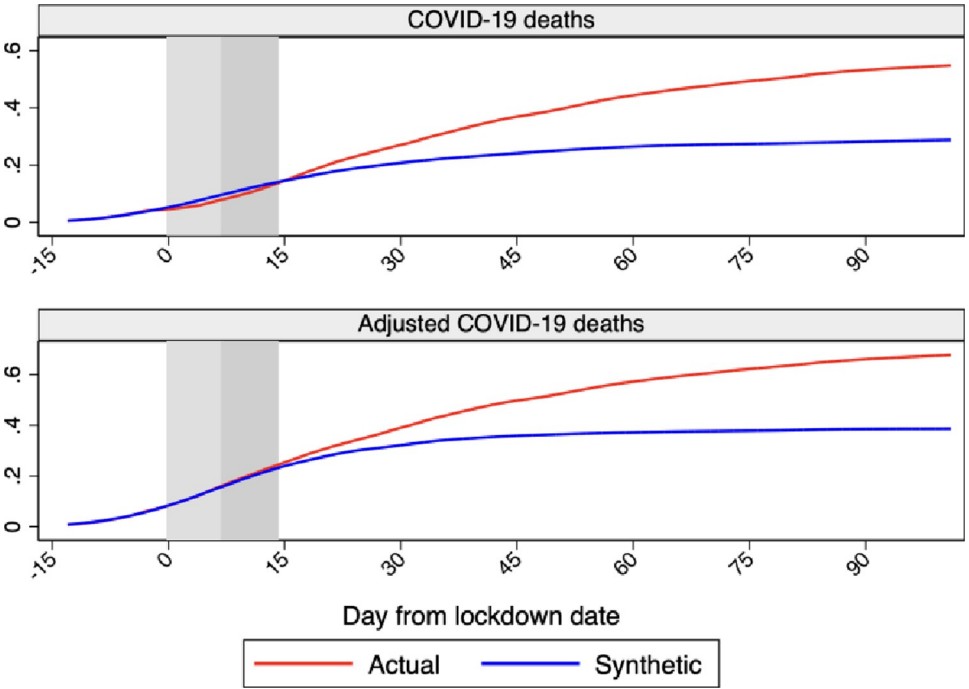

**Fig 4. Profiles of cumulative COVID-19 deaths and COVID-19 adjusted deaths for Sweden and synthetic Sweden.**
*Notes*: The profiles of cumulative COVID-19 deaths and COVID-19 adjusted deaths for Sweden and synthetic Sweden are shown in the figure. Horizontal axis measures days since the lockdown start that is normalized at day 0. The red line shows the profile for Sweden and the blue line shows the profile for synthetic Sweden. The vertical bands indicate the first 14 days after the lockdown start, with the lightest color as the first 7 and the darker as additional 7 days. Data source: Our World in Data and Financial Times.

the profile of adjusted COVID-19 deaths doesn't look much different. The only change is that the profiles of Sweden and synthetic Sweden start to diverge much earlier than before suggesting a faster effect of the lockdown. However, due to the high variance in the median lag between infections and deaths, COVID-19 deaths are not a reliable measure of the delay of the lockdown.

**5.2.2. How long does it take for the lockdown to show its effects?.** To understand the delay with which the lockdown displays its effects we turn to infections, the measure on which previous work has focused the most. Born et al. [1] and Cho [2] found that the effects of the lockdown would occur with a delay of three to five weeks after its implementation.

Fig 6 compares the cumulative infections and the cumulative positive rate. While the profiles of COVID-19 infections of Sweden and synthetic Sweden start to diverge about 20 days after the lockdown implementation, more or less as in Born et al. [1] and Cho [2], the positive rate of Sweden jumps above synthetic Sweden already after about 7 days. This suggests that the observed delay in the cumulative number of infections is in large part artificially generated by the extremely slow testing rate in Sweden during the first phase of the epidemic that we documented in Section 2.2, which is "filtered away" using the ratio between the number of infections and the number of tests.

Fig 7 presents the profile of Sweden versus synthetic Sweden for all outcomes considered over our sample period, in terms of both daily and cumulative values. The daily indicators consistently show an even faster robust effect of the lockdown taking place between few days and a week after its introduction.

Taken together, our multiple indicators show that a lockdown would have had effects after about a week, in line with previous studies that, using different methodologies or focusing on

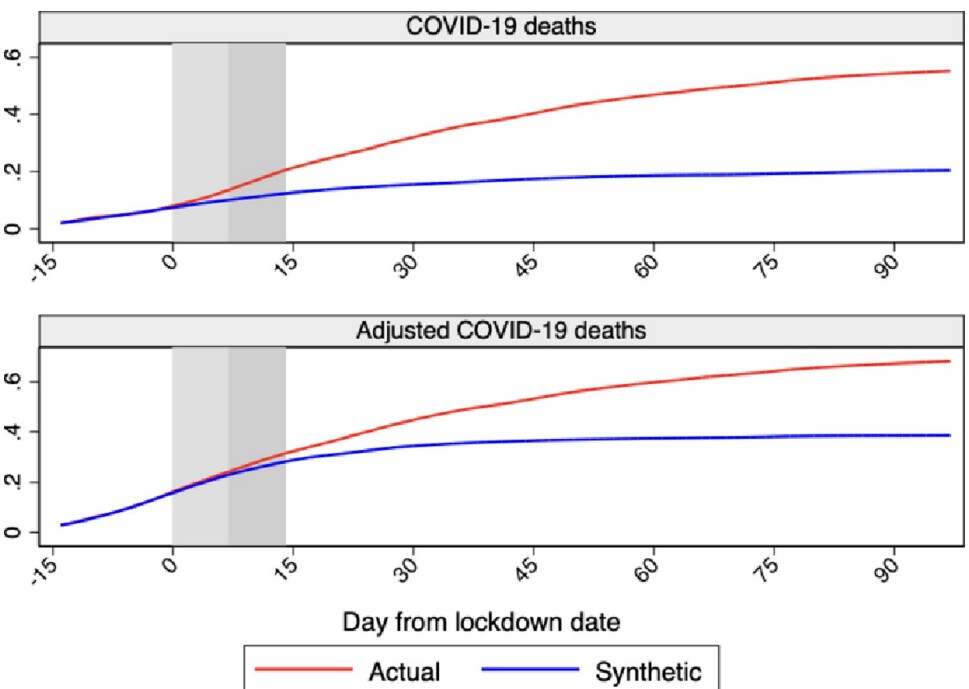

**Fig 5. Profiles of cumulative COVID-19 deaths and COVID-19 adjusted deaths for Sweden and synthetic Sweden shifting the treatment date one week further.** *Notes*: The profiles of cumulative COVID-19 deaths and COVID-19 adjusted deaths for Sweden and synthetic Sweden shifting the treatment date one week further are shown in the figure. Horizontal axis measures days since the lockdown start that is normalized at day 0. The red line shows the profile for Sweden and the blue line shows the profile for synthetic Sweden. The vertical bands indicate the first 14 days after the lockdown start, with the lightest color as the first 7 and the darker as additional 7 days. Data source: Our World in Data and Financial Times.

different countries, found considerable effects of the lockdown already a few days after its implementation (e.g., [4–6]). Friedson et al. [26] estimate the effect of the lockdown in California with the synthetic control methodology and find that the rate of growth in California's COVID-19 cases was substantially lower relative to the synthetic control just four days after the lockdown implementation. The much longer delay suggested by the cumulative infections indicator here and in Born et al. [1] and Cho [2], could have been a natural effect of the inertia intrinsic to stock rather than flow measures. In the case of Sweden, it could also be generated by the extremely low rate of testing that Sweden maintained in the first part of our sample period, followed by a strong increase in the last part.

The inclusion of the positive rate among our outcomes allows us to shed light on the relative importance of these two possible explanations. Fig 7 shows a rather small additional delay in the observed effect of the lockdown on the cumulative positive rate relative to the daily indicator. This suggests that the delay in the effect of the lockdown on cumulative infections is almost entirely driven by the changes in Swedish testing policy during our period, which are filtered out when using the positive rate.

**5.2.3. Quantitative effects of the lockdown.** Visual inspection of Fig 7 shows that each of our indicators consistently suggests that a lockdown would have had a strong effect in reducing the impact of COVID-19 in Sweden.

Fig 8 shows the percentage differences between synthetic Sweden and actual Sweden. Quantitatively, our estimates of the effects of a lockdown are often somewhat higher than previous works. Starting with cumulative COVID- 19 infections–not our preferred outcome in the light of the evidence in Section 2.2 –we estimate a 61% reduction by May 17, 2020 (when

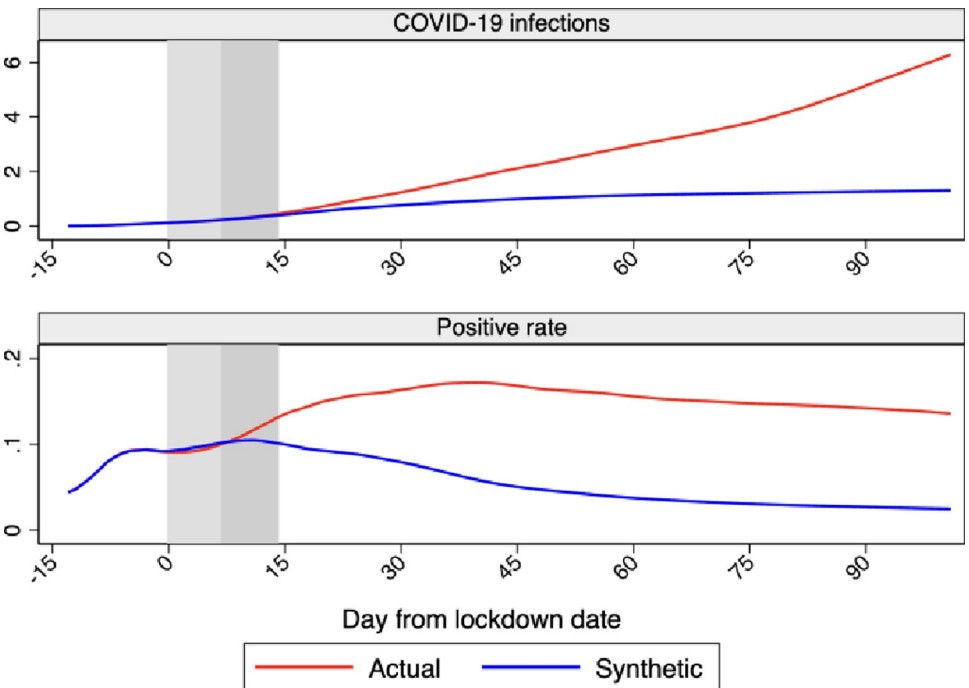

**Fig 6. Profiles of cumulative infections and positive rate for Sweden and synthetic Sweden.** *Notes*: The profiles of cumulative infections and positive rate for Sweden and synthetic Sweden are shown in the figure. Horizontal axis measures days since the lockdown start that is normalized at day 0. The red line shows the profile for Sweden and the blue line shows the profile for synthetic Sweden. The vertical bands indicate the first 14 days after the lockdown start, with the lightest color as the first 7 and the darker as additional 7 days. Data source: ECDC and Our World in Data.

Born et al. [1] estimate a reduction of 48%), and a 71% reduction by June 7, 2020 (when Cho [2] estimates a reduction of 75%). We then estimate a 40% reduction in cumulative COVID-19 deaths by May 17, 2020 (when Born et al. [1] estimate a 34% reduction) and a 41% reduction in cumulative adjusted COVID-19 deaths by June 13, 2020 (when Cho [2] finds a 25% reduction in excess deaths). On June 30, 2020, the end of our sample period, the reduction in cumulative adjusted COVID-19 deaths is 4 percentage points lower than for cumulative COVID-19 deaths (-43% vs. -47%), which is consistent with our conjecture that Sweden had a rather encompassing approach when assigning deaths to COVID-19.

## 5.3 Robustness checks

Since choices regarding the key elements of the synthetic control method listed in Section 3.2 are somewhat "ad hoc", in this section we briefly present the results of a number of robustness checks.

We consider four cases and compare the results with those from the baseline case presented in Section 5.2. The four cases considered are obtained by varying, one at the time, the set of countries in the donor pool (Cases 1 and 2), the treatment date (Case 3), and the set of country characteristics (Case 4). Detailed tabulations for each of the four cases are available upon request, while the percentage differences between synthetic Sweden and Sweden for each case are shown in Fig 9 with reference to the cumulative outcomes.

Case 1 includes France and Spain ignoring the presence of negative values of daily infections and daily deaths for these two countries. This makes the results for this case more comparable with those in [1]. The differences with respect to those in Section 5.2 are only minor.

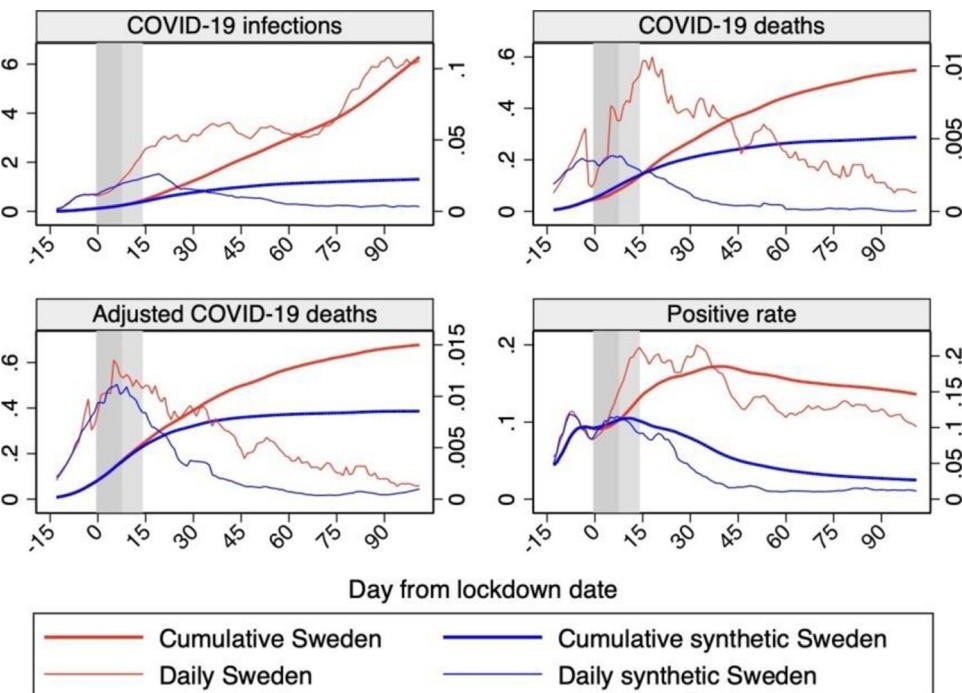

**Fig 7. Profiles of daily and cumulative outcomes for Sweden and synthetic Sweden.** *Notes*: The profiles of daily and cumulative outcomes for Sweden and synthetic Sweden are shown in the figure. Horizontal axis measures days since the lockdown start that is normalized at day 0. The red line shows the profile for Sweden and the blue line shows the profile for synthetic Sweden. The vertical bands indicate the first 14 days after the lockdown start, with the lightest color as the first 7 and the darker as additional 7 days. Data source: ECDC, Our World in Data, and Financial Times.

Case 2 expands the donor pool to include most of the countries considered by Cho [2]. This makes the results for this case more comparable with his results. Expanding the donor pool in this way does not affect the results for COVID-19 deaths and the positive rate. When looking at cumulative COVID-19 infections and cumulative adjusted COVID-19 deaths, the profiles for synthetic Sweden are higher than in the baseline case. When looking at the percentage difference, we see that the percentage difference for the case of cumulative COVID-19 deaths is higher than for COVID-19 adjusted deaths, the opposite than our baseline case.

Case 3 shifts the treatment date 7 days further to account for the average length of the incubation period. Again, results hardly change.

Case 4 adds to the population size and the share of urban population other economic and socio-demographic indicators (average household size, median age, share of people aged 70+, life expectancy, GDP per capita, and hospital beds per thousands). Adding all these controls hardly changes our results.

## 6 Discussion and conclusions

In this paper, we compare several indicators of the spread and consequences of the COVID-19 pandemic that are often used in isolation for both cross-country comparisons and policy evaluation. We focus on the highly debated case of Sweden–to our knowledge the only country with good data that did not impose a lockdown during the first wave of the pandemic in Spring 2020. We construct counterfactuals for what would have happened if Sweden had imposed such a lockdown using the synthetic control methodology, specifically optimized for each of the outcomes considered.

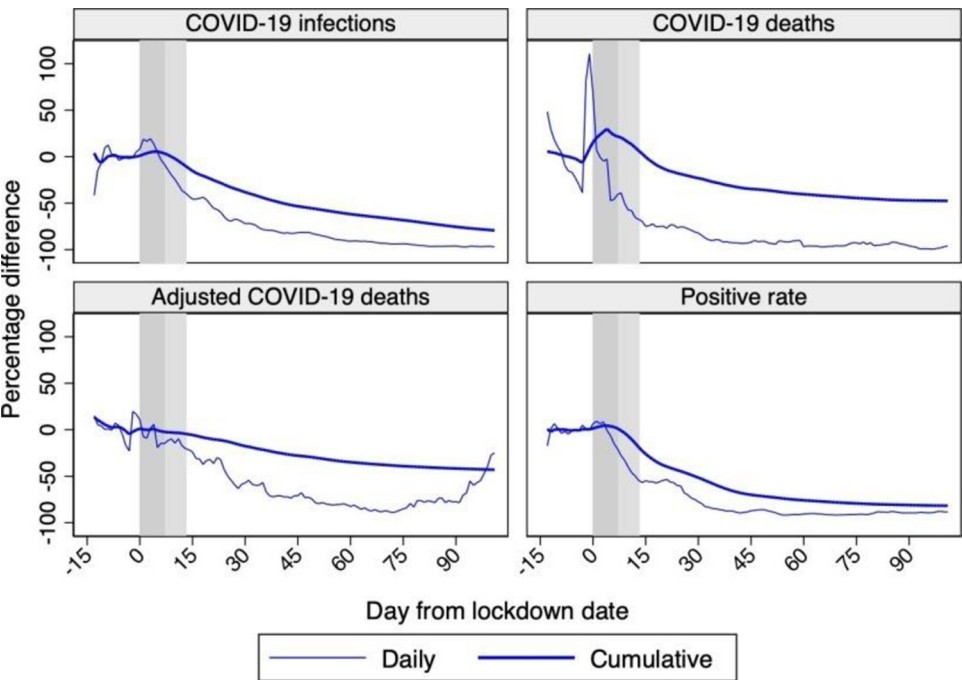

**Fig 8. Percentage differences between synthetic Sweden and Sweden.** *Notes*: The percentage differences between synthetic Sweden and Sweden in terms of COVID-19 infections, deaths, adjusted deaths, and the positive rate are shown in the four panels. Horizontal axis measures days since the lockdown start that is normalized at day 0. The vertical bands indicate the first 14 days after the lockdown start, with the lightest color as the first 7 and the darker as additional 7 days. Data source: ECDC, Our World in Data, and Financial Times.

We address several problems in the data that had not been previously identified, and we propose a novel methodology that uses weekly data on excess mortality to correct the daily series of total COVID-19 deaths for under-reporting and cross-country heterogeneity in the definition and measurement of deaths.

All our indicators suggest that a lockdown would have had a strong effect in reducing the impact of COVID-19 in Sweden. Most importantly for the design and timing of future policies, we study the cumulative positive rate and four additional daily indicators, finding that a lockdown would have had a sizable effect already within one week after its introduction. The much longer delay estimated in previous studies focusing on the number of COVID-19 infections appears to result from the extremely low frequency of testing that occurred in this country in early Spring 2020 followed by a sustained increase in late Spring and early Summer.

Our study highlights the importance of looking at multiple indicators when evaluating policies or comparing countries. It also highlights the need of improving the quality of available data. The best way to produce comparable indicators for policy evaluation would of course be to have more homogeneous statistics over time and across countries, possibly at a finer geographical level within each country.

Our results do not imply that a lockdown would have been optimal or efficient for Sweden, as the very high costs of a lockdown should also be considered. Future work should address these important, complementary aspects necessary for a proper cost-benefit analysis.

Lastly, several country-specific factors, such as demographic structure, socio-economic characteristics and, lifestyle, are important determinants of the dynamics of the epidemic, and our results on Sweden do not imply that a lockdown would have had the same effect in another country.

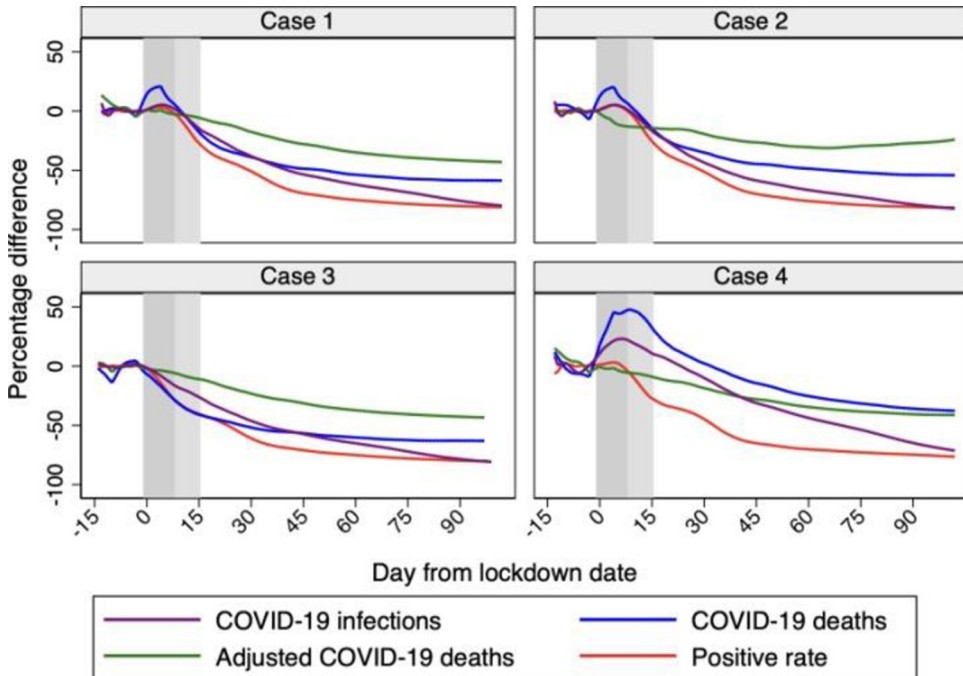

**Fig 9. Percentage differences in cumulative outcomes between synthetic Sweden and Sweden.** *Notes*: The percentage differences between synthetic Sweden and Sweden in terms of COVID-19 infections, deaths, adjusted deaths, and the positive rate are shown in the four panels. The top left panel are the results for case 1 (expanded donor pool with France and Spain). The top right panel are the results for case 2 (expanded donor pool with other European countries). The bottom left panel are the results for case 3 (shifting the treatment date). The bottom right panel are the results for case 4 (adding extra control variables). Horizontal axis measures days since the lockdown start that is normalized at day 0. The vertical bands indicate the first 14 days after the lockdown start, with the lightest color as the first 7 and the darker as additional 7 days. Data source: ECDC, Our World in Data, and Financial Times.

A practical implication of our study is that, when planning a lockdown, authorities should know that its effects will start already after a few days, rather than after several weeks as argued by previous studies. Another one is that to have a full understanding of the state of an epidemic, all available indicators must be considered.

A limitation of our study is that it cannot be replicated for most other countries for which data are available, as they introduced a lockdown very early on. To our knowledge, the United States is the only country for which studies similar to our own have been performed [4–6]. These studies also find short delays in the effects of a lockdown, suggesting that our findings are not unique to Sweden.

## Author Contributions

**Conceptualization:** Chiara Latour, Franco Peracchi, Giancarlo Spagnolo.

**Data curation:** Chiara Latour, Franco Peracchi, Giancarlo Spagnolo.

**Formal analysis:** Chiara Latour, Franco Peracchi, Giancarlo Spagnolo.

**Methodology:** Chiara Latour, Franco Peracchi, Giancarlo Spagnolo.

**Writing – original draft:** Chiara Latour, Franco Peracchi, Giancarlo Spagnolo.

**Writing – review & editing:** Chiara Latour, Franco Peracchi, Giancarlo Spagnolo.

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
