## [Decision Letter · Decision Letter 0]

10 Sep 2021

PONE-D-21-23290Assessing Alternative Indicators for Covid-19 Policy Evaluation, with a Counterfactual for Sweden.PLOS ONE

Dear Dr. SPAGNOLO,

Thank you for submitting your manuscript to PLOS ONE. After careful consideration, we feel that it has merit but does not fully meet PLOS ONE’s publication criteria as it currently stands. Therefore, we invite you to submit a revised version of the manuscript that addresses the points raised during the review process.

We look forward to receiving your revised manuscript.

Kind regards,

Bing Xue, Ph.D.

Academic Editor

PLOS ONE

Journal Requirements:

2. Please update your submission to use the PLOS LaTeX template. The template and more information on our requirements for LaTeX submissions can be found at http://journals.plos.org/plosone/s/latex

Reviewers' comments:

Reviewer's Responses to Questions

**Comments to the Author**

1. Is the manuscript technically sound, and do the data support the conclusions?

Reviewer #1: Yes

Reviewer #2: Partly

Reviewer #3: Yes

2. Has the statistical analysis been performed appropriately and rigorously? 

Reviewer #1: Yes

Reviewer #2: Yes

Reviewer #3: Yes

3. Have the authors made all data underlying the findings in their manuscript fully available?

Reviewer #1: No

Reviewer #2: No

Reviewer #3: Yes

4. Is the manuscript presented in an intelligible fashion and written in standard English?

Reviewer #1: Yes

Reviewer #2: No

Reviewer #3: Yes

5. Review Comments to the Author

Reviewer #1: 1. Abstract and introduction section has written clearly.

2. Section '2.2 Heterogeneity over time', please report the sources of the following statement, .... "Prior to March 12, 2020, the Swedish strategy was to test all people who had been in areas

considered at risk, like China and Italy, but due to shortages in testing equipment, this strategy was rapidly

changed, and testing was only targeted towards people with heavy symptoms and in need of hospitalization

and medical care staff."

"Sweden then reversed this policy announcing on March 31, 2020, a plan to expand

testing capacity to all critical services: the aim was to carry out 100,000 tests per week."

3. Section 'Methodology'. The author(s) has clearly explained the synthetic control method.

4. section '4 Data' data analysis has been explained clearly.

5. Section '5 Results' has discussed clearly and supported with previous literature.

6. Please include the practical and managerial implication section before conclusion

7. Please include the limitation and future study, and highlight the impacts/values of this study in the society.

Reviewer #2: In this article, the authors utilized the synthetic control method to study the following questions: What would happen if the Swedish government adopted an approach of lockdown to prevent the spread of the pandemic after the first wave of COVID-19's outbreak? By using multiple indicators, the authors showed that a lockdown would have displayed its effects within a week from its introduction. Their results on Sweden suggest that a lockdown was not optimal or efficient owing to the very high costs of a lockdown. In addition, the authors proposed a methodology that used weekly data on excess mortality to correct the daily series of total COVID-19 deaths for under-reporting and cross-country heterogeneity in the definition and measurement of deaths.

The paper has its merits, but on the whole, the paper has not been well written. It's very hard for me to read this article. The main problems of this work are as follows.

First, the effect of lockdown against the spread of the COVID-19 pandemic depends on many factors, including lifestyle of people in different countries, the density of population, sizes of cities, time of taking lockdown measure, and so on. All this kinds of problems were not taken into account or discussed in this paper. Because of the above problems, it is very doubtful whether the conclusion of the paper is general.

Second, eight indicators were employed in this studies. However, the authors did not discuss the relationships between these indicators. In data analysis, adding an indicator will bring information and noise at the same time. When using multiple indicators for quantitative analysis, you must pay attention to the orthogonality and dimensional consistency between indicators. In addition, the effectiveness of indicators and the redundancy of indicator system should be discussed.

Third, some places are unclear. For example, there are no substantial differences between the titles and notes of Figures 4, 5 and 6.

Fourth, the paper is poorly written by an author whose first language is clearly not English. In fact, there are too many problems in the grammar, usage, and overall readability of the manuscript. The manuscript should be revised to fix the grammatical errors and improve the overall readability of the text before it is accepted for possible publication. I suggest the author(s) have a fluent, preferably native, English-language speaker thoroughly copyedit this manuscript for language usage, spelling, and grammar.

Reviewer #3: Summary

The paper studies whether Sweden would have had less COVID-19 infections and (adjusted) COVID-19 related deaths if authorities had introduced a lockdown in mid-March 2020, i.e. at the beginning of the first wave, following the example of other western economies. It does so by applying the synthetic control method, following the example of Born et al. (2020) and Cho (2020 – with the latter paper “in terms of scope and methodology … closest in spirit” to Born et al. 2020). It qualitatively confirms the results of those studies but highlights that after correcting for data problems and accounting more properly for the low testing frequency in Sweden at the beginning of the pandemic the negative impact of a Swedish lockdown on infections in the country would have been even more swift than what has been found by Born et al (2020) and Cho (2020).

Comments

1. The core of the paper is not reflected in its title which suggests – as also mentioned in the first sentence of the introduction that – that it has a much broader focus, namely to analyze “what indicators should be used to monitor epidemics and evaluate policies, such as lockdowns or other nonpharmaceutical interventions.” However, the latter question is not addressed in the paper. For sure, in section 4 the authors introduce the “adjusted COVID-19 deaths” indicator. Moreover, somewhat related, but already very much focused on Sweden, the paper discusses the relationship between the number of reported infections and testing intensity which leads to the identification of the “positive rate” as another indicator to measure the impact of NPIs and lockdowns. However, the indicator itself has been widely discussed in the literature before (see e.g. Hasell et al. (2020), A cross-country database of COVID-19 testing, Scientific data, 7(1), 1-7). More importantly, I do not find supporting evidence for the claim that “looking at several indicators”, including adjusted deaths and the positive rate, is needed to “derive robust conclusions” (on what? on the COVID-19 effects of lockdowns?), also because the analysis is just a case study of Sweden. Thus, I tend to disagree with conclusion that the “study highlights the importance of looking at multiple indicators when evaluating policies or comparing countries” (page 17). Accordingly, I recommend that the authors change the title and the introduction / conclusion accordingly, focus on the Swedish case and refrain from making general statements suggesting that the paper is about the pros and cons of several indicators measuring the state of a pandemic.

2. With the core of the paper being the Swedish case, the paper basically aims at reconsidering the results obtained by Born et al. (2020) and Cho (2020) with some new indicators (the “adjusted COVID-19 deaths” and the “positive rate”) as well as cumulative versus daily data. Indeed, most of the paper can be seen as a kind of robustness check and extension of Born et al. (2020) and Cho (2020). It does a good job in doing so and in qualitative terms confirms the results of previous studies. Thus, I recommend more restraint when discussing the novelty of the results (this holds even more when considering the 2021 version of Born et al. (2020) published in PLOS One https://journals.plos.org/plosone/article?id=10.1371/journal.pone.0249732 given the extensions, also in terms of the post-treatment period considered, compared to the 2020 version). Indeed, for death indicators the results seem to be broadly the same as in Born et al. (2020) and Cho (2020). Things are different for infections. However, even here the paper’s main new insight – that a Swedish lockdown would have worked faster than previous research suggests – is mainly driven by the positive rate and when making use of daily instead of cumulative cases.

3. The paper develops the “adjusted COVID-19 deaths” indicator. As shown in section 4.3 it adds to the number of reported COVID-19 deaths excess mortality, if excess mortality is positive. The paper argues that such an adjustment is needed as “the number of reported COVID-19 deaths is likely to represent a downward biased estimate of total COVID-19 deaths” (p. 9). However, I do not find arguments why this should be the case. Indeed, on pages 2 and 3 arguments are presented suggesting that reported COVID-19 cases might be overreported (with Sweden itself being a candidate as it “has been rather generous in ascribing deaths to COVID-19” (p. 3). Moreover, excess mortality is unlikely to be affected by COVID-19 deaths only (as stated in the form of an assumption on page 9) but also by nonpharmaceutical interventions themselves which can raise but also lower mortality rates independently from reported COVID-19 deaths. Moreover, these effects are likely to vary across countries, for example due to different demographic characteristics. Thus, I have doubts whether the claim made on the bottom of page 8 is correct, namely that “excess mortality is … robust to structural differences across countries.” Accordingly, I would welcome a more balanced discussion about the advantages and disadvantages of the adjusted compared to unadjusted COVID-19 deaths.

Minor points

- the direct quotes on page 3 are pretty lengthy and could be summarized by the authors in a shorter way.

- Figures are presented without reference to data sources. This should be corrected.

- Figure 1: Typo: inhabitants

- Page 5, second para, first sentence: Typo: “Sweden finally managed to step up …

- Page 5: it is not clear to me why the risk of overfitting may increase with the size of the donor pool, especially when T0 is small (I assume T0 refers to the length of the pre-intervention period).

Page 10, footnote 11: Please clarify whether the footnote refers to the baseline, the robustness checks or both.

Page 10, footnote 13: The message given is unclear without having read Cho (2020)

Page 11, Section 5.2, second para: Is “overreporting” the only reason for Sweden showing a slightly smaller number of COVID-19 deaths than its synthetic counterpart?

Page 12, second para, last sentence: Is the variance argument not implying that COVID-19 deaths are in general not a useful measure for assessing the speed of the impact of NPIs and lockdowns?

Page 12, last para: the sentence “While the curves … as in Born et al. (2020) and Cho (2020)” does not seem to have a main clause.

Page 14, last para: are the reported differences between results obtained in this paper compared to Born et al. (2020) and Cho (2020) significant?

6. PLOS authors have the option to publish the peer review history of their article (what does this mean?). If published, this will include your full peer review and any attached files.

Reviewer #1: No

Reviewer #2: No

Reviewer #3: No

---

## [Author Response · Author response to Decision Letter 0]

19 Oct 2021

We responded to reviewers comments in the attached file "Response to reviewers".

---

## [Decision Letter · Decision Letter 1]

24 Jan 2022

PONE-D-21-23290R1Assessing Alternative Indicators for Covid-19 Policy Evaluation, with a Counterfactual for Sweden.PLOS ONE

Dear Dr. SPAGNOLO,

Thank you for submitting your manuscript to PLOS ONE. After careful consideration, we feel that it has merit but does not fully meet PLOS ONE’s publication criteria as it currently stands. Therefore, we invite you to submit a revised version of the manuscript that addresses the points raised during the review process.

We look forward to receiving your revised manuscript.

Kind regards,

Bing Xue, Ph.D.

Academic Editor

PLOS ONE

Journal Requirements:

Reviewers' comments:

Reviewer's Responses to Questions

**Comments to the Author**

1. If the authors have adequately addressed your comments raised in a previous round of review and you feel that this manuscript is now acceptable for publication, you may indicate that here to bypass the “Comments to the Author” section, enter your conflict of interest statement in the “Confidential to Editor” section, and submit your "Accept" recommendation.

Reviewer #1: All comments have been addressed

Reviewer #2: All comments have been addressed

Reviewer #3: All comments have been addressed

2. Is the manuscript technically sound, and do the data support the conclusions?

Reviewer #1: Yes

Reviewer #2: Yes

Reviewer #3: Yes

3. Has the statistical analysis been performed appropriately and rigorously? 

Reviewer #1: Yes

Reviewer #2: Yes

Reviewer #3: Yes

4. Have the authors made all data underlying the findings in their manuscript fully available?

Reviewer #1: No

Reviewer #2: No

Reviewer #3: Yes

5. Is the manuscript presented in an intelligible fashion and written in standard English?

Reviewer #1: Yes

Reviewer #2: Yes

Reviewer #3: Yes

6. Review Comments to the Author

Reviewer #1: The author has improved the paper accordingly but there is still lack of information about the practical contribution and impact of the finding in the society. The paper needs to explain some potential limitation before final accepting the paper for publishing.

Reviewer #2: Compared with the previous version, the quality of this papers has been improved. The authors tried their best to address my comments.

Reviewer #3: Dear authors,

thank you for the revised version of the paper which takes on board most of my concerns I raised in the first review. Only a few minor points are left:

Page 1, third para, line 8. Typo: “… the number of recorded infections to the number of tests …”

Page 1, fourth para, line 4. Suggestion for using a less strong language, i.e. something like: “This finding contrasts with the longer lags (three to five weeks) in the effect of lockdowns estimated in Born et al. (2020) and Cho (2020) …

Page 1, fourth para, last line: Which are the other studies you have in mind showing results which are more consistent with you study?

Page 4, Heterogeneity over time: This subsection, as it stands, only deals with testing policy (Is this the only heterogeneity over time?). If this remains the case the subtitles should reflect that “2.1 Heterogeneity across countries” deals with various issues (ascribing COVID deaths and testing), while 2.2. only deals with testing.

Page 4, second para: who are the “people who had been in areas considered at risk”? Could you be a bit more explicit?

Page 4, third para: Wording suggestion “… those worried about their infection states but unable to be tested before, hence exhibited a higher probability ….”

Page 7, second para: Wording suggestion: “… argue that excess mortality is more comparable across countries because it is less sensitive …”

Page 10, last para, second line: Suggestion for toning down language: “… our estimates of the effects of a lockdown are often somewhat higher than in previous works.” (there is one case, where your effect is less strong than in Cho (2020). Moreover, given that tests of significance cannot be run, I think that introducing “somewhat” is a way of expressing that the difference between numbers like 34 and 40% (cumulative COVID deaths) is not that large.

7. PLOS authors have the option to publish the peer review history of their article (what does this mean?). If published, this will include your full peer review and any attached files.

Reviewer #1: No

Reviewer #2: No

Reviewer #3: No

---

## [Author Response · Author response to Decision Letter 1]

6 Feb 2022

Review Comments to the Author

Reviewer #1: The author has improved the paper accordingly but there is still lack of information about the practical contribution and impact of the finding in the society. The paper needs to explain some potential limitation before final accepting the paper for publishing.

We have added the practical implications and limitations of our study in the last two paragraphs of the conclusions. 

Reviewer #3: Dear authors,

thank you for the revised version of the paper which takes on board most of my concerns I raised in the first review. Only a few minor points are left:

Page 1, third para, line 8. Typo: “… the number of recorded infections to the number of tests …” 

Corrected 

Page 1, fourth para, line 4. Suggestion for using a less strong language, i.e. something like: “This finding contrasts with the longer lags (three to five weeks) in the effect of lockdowns estimated in Born et al. (2020) and Cho (2020) … 

Corrected 

Page 1, fourth para, last line: Which are the other studies you have in mind showing results which are more consistent with you study? 

Corrected 

Page 4, Heterogeneity over time: This subsection, as it stands, only deals with testing policy (Is this the only heterogeneity over time?). If this remains the case the subtitles should reflect that “2.1 Heterogeneity across countries” deals with various issues (ascribing COVID deaths and testing), while 2.2. only deals with testing. Corrected 

Page 4, second para: who are the “people who had been in areas considered at risk”? Could you be a bit more explicit? 

Corrected 

Page 4, third para: Wording suggestion “… those worried about their infection states but unable to be tested before, hence exhibited a higher probability ….” 

Corrected 

Page 7, second para: Wording suggestion: “… argue that excess mortality is more comparable across countries because it is less sensitive …” 

Corrected

Page 10, last para, second line: Suggestion for toning down language: “… our estimates of the effects of a lockdown are often somewhat higher than in previous works.” (there is one case, where your effect is less strong than in Cho (2020). Moreover, given that tests of significance cannot be run, I think that introducing “somewhat” is a way of expressing that the difference between numbers like 34 and 40% (cumulative COVID deaths) is not that large. 

Corrected

Additional comment: 

Regarding the answers NO to the question whether we made data available.

The sources of the data are described in Section 4 of our paper and are the following:

Our World in Data - Covid deaths and number of infections

European Centre for Disease Prevention and Control ECDC - Covid tests

Financial Times - excess mortality

---

## [Editor Report · Decision Letter 2]

17 Feb 2022

Assessing Alternative Indicators for Covid-19 Policy Evaluation, with a Counterfactual for Sweden.

PONE-D-21-23290R2

Dear Dr. SPAGNOLO,

We’re pleased to inform you that your manuscript has been judged scientifically suitable for publication and will be formally accepted for publication once it meets all outstanding technical requirements.

Kind regards,

Bing Xue, Ph.D.

Academic Editor

PLOS ONE
---

## [Editor Report · Acceptance letter]

4 Mar 2022

PONE-D-21-23290R2 

Assessing Alternative Indicators for Covid-19 Policy Evaluation, with a Counterfactual for Sweden∗ 

Dear Dr. Spagnolo:

I'm pleased to inform you that your manuscript has been deemed suitable for publication in PLOS ONE. Congratulations! Your manuscript is now with our production department. 

Kind regards, 

on behalf of

Professor Bing Xue 

Academic Editor

PLOS ONE